# Spatiotemporal Interaction and Socioeconomic Determinants of Rural Energy Poverty in China

**DOI:** 10.3390/ijerph191710851

**Published:** 2022-08-31

**Authors:** Siyou Xia, Yu Yang, Xiaoying Qian, Xin Xu

**Affiliations:** 1Key Laboratory of Regional Sustainable Development Modeling, Institute of Geographic Sciences and Natural Resources Research, Chinese Academy of Sciences, Beijing 100101, China; 2College of Resources and Environment, University of Chinese Academy of Sciences, Beijing 100049, China; 3Institute of Strategy Research for Guangdong-Hong Kong-Macao Greater Bay Area, Guangzhou 510070, China; 4Population Research Institute, Nanjing University of Posts and Telecommunications, Nanjing 210042, China

**Keywords:** rural energy poverty, spatiotemporal interaction, socioeconomic determinants, collaborative poverty reduction, China

## Abstract

This study investigated the energy poverty spatiotemporal interaction characteristics and socioeconomic determinants in rural China from 2000 to 2015 using exploratory time–space data analysis and a geographical detector model. We obtained the following results. (1) The overall trend of energy poverty in China’s rural areas was “rising first and then declining”, and the evolution trend of energy poverty in the three regions formed a “central–west–east” stepwise decreasing pattern. (2) There was a dynamic local spatial dependence and unstable spatial evolution process, and the spatial agglomeration of rural energy poverty in China had a relatively higher path dependence and locked spatial characteristics. (3) The provinces with negative connections were mainly concentrated in the central and western regions. Anhui and Henan, Inner Mongolia and Jilin, Jilin and Heilongjiang, Hebei and Shanxi, and Liaoning and Jilin constituted a strong synergistic growth period. (4) From a long-term perspective, the disposable income of rural residents had the greatest determinant power on rural energy poverty, followed by per capita GDP, rural labor education level, regulatory agencies, and energy investment. In addition, our findings showed that the selected driving factors all had enhanced effects on rural energy poverty in China through interaction effects.

## 1. Introduction

Energy poverty is a huge global challenge [1,2,3]. According to the data released by the World Bank in 2020, nearly 1 billion people around the world live in energy-impoverished conditions and rely on traditional biomass such as wood, agricultural residues, and animal dung for cooking [4], which poses a great threat to sustainable development, health, and education [5]. In recent years, this issue has attracted considerable attention from academia, governments, and international organizations. The United Nations put forward the “Sustainable Energy for All” initiative in 2012, calling on all countries in the world to act together to ensure that everyone enjoys modern, clean, and efficient living energy, and to jointly deal with energy poverty [6]. In 2015, the Sustainable Development Goals (SDGs) were formulated by the United Nations, among which SDG7 clearly proposes to ensure that all people have access to affordable, reliable, sustainable, and modern energy [7].

Most existing studies confirm that energy poverty harms social and economic wellbeing [8,9,10,11], causes education and gender inequality [12,13,14,15], increases residents’ health risk [16,17,18,19,20], and leads to ecological degradation [21,22,23]. Research on the influencing factors of energy poverty shows that household income, the educational level of residents, and social factors (age, ability, ethnicity) are the main causes of energy poverty [24,25,26,27,28,29,30,31,32]. With the continuous promotion of energy transitions, the impact of low-carbon energy transition on energy poverty has also received extensive attention. Many studies have shown that the transition from fossil fuels to renewable energy plays an important role in alleviating energy poverty [33,34,35,36]. However, some scholars have pointed out that energy transition will increase the energy use cost of residents to a certain extent, bring additional burdens to low-income families, and aggravate energy poverty [37,38,39,40]. Although a great number of studies have been conducted on energy poverty, two limitations still exist. First, some previous studies have investigated the spatial pattern of energy poverty using exploratory spatial data analysis (ESDA); however, ESDA only targets cross-sectional data and ignores the temporal and spatial dynamics of rural energy poverty. In fact, rural energy poverty is a complex process of space–time change, which will vary with changes in time and space. This article introduces the exploratory time–space data analysis (ESTDA) framework to study the spatiotemporal interaction characteristics of rural energy poverty in China [41]. Importantly, the ESTDA framework can integrate time and space to study spatiotemporal interaction as well as to compensate for the shortcomings of ESDA, which ensures the accuracy of the estimation results [42]. Second, previous studies have generally focused on the impact of a single factor, such as household income, energy prices, buildings, and equipment, on energy poverty. However, rural energy poverty is complex and may be affected by many factors. It is urgent to explore the interaction of multiple factors on energy poverty. Therefore, this study involved a single factor and interactive analysis on the socioeconomic factors of rural energy poverty to better understand the spatiotemporal dynamics and socioeconomic determinants of rural energy poverty in China.

More surprisingly, as the largest developing country in the world, China’s energy poverty problem is more serious and complicated than other countries. Problems such as low energy consumption level, unreasonable energy consumption structure, and weak energy utilization capacity are more prominent. Especially in China’s rural areas, where rural families are mainly reliant on traditional biomass energy, a large proportion of the population has difficulty accessing and using modern forms of energy [6]. According to the 2010 national census data, up to 76% of rural households in China use coal and firewood as their main cooking fuels [2], and are without the benefit of stable electricity and other clean energy as their main living energy, which seriously limits the realization of the goal of energy poverty alleviation in rural China. Meanwhile, due to the gap between urban and rural households, compared with urban areas, China’s rural areas not only lack clean and efficient energy for household use [43], but also lack the ability to pay for modern energy, and the energy consumption structure remains characterized by high carbonization and non-cleaning [6]. Therefore, in the context of energy transition and sustainable development, the means by which to better solve the problem of rural energy poverty in China has become a hot research topic.

Therefore, the objectives of this paper are: (1) to investigate the spatiotemporal interaction characteristics of rural energy poverty in China; and (2) to study the impact of socioeconomic factors on rural energy poverty, with the aim of providing a reference for risk identification and the prevention of energy poverty.

## 2. Data and Methodology

### 2.1. Indicators and Data Sources

An accurate assessment of energy poverty is essential to guide policy development and implementation [5,44]. Energy poverty is widely understood as the absence of sufficient choice in accessing adequate, affordable, reliable, high-quality, safe, and environmentally friendly electricity, clean fuel, and modern energy services to support socioeconomic and human development [45]. Therefore, energy access and energy services are the core of energy poverty. Energy access refers to the provision of electricity and other modern energy services for all; modern energy services specifically refer to household access to electricity, the use of clean kitchenware, and the elimination of fuels (such as traditional biomass energy and coal) and stoves that cause indoor air pollution. In view of this, the rural energy poverty index used in this paper was released by the *Journal of Global Change Data & Discovery* [46]. The index was constructed by Zhao et al. from the two dimensions of energy access and energy services (Table 1), was calculated by the weighted summation method [6], and can comprehensively reflect the situation of energy poverty in rural China. This study used this index to study the spatiotemporal interaction characteristics and socioeconomic determinants of rural energy poverty in China. The data names, descriptions, and sources are shown in Table 1. Due to data unavailability, Tibet, Hong Kong, Macao, and Taiwan of China were excluded from the scope of this study.

### 2.2. ESTDA Analysis Framework

The ESTDA analysis framework was used to analyze the spatiotemporal dynamic characteristics of rural energy poverty in China, including the local indicators of spatial association (LISA) time path, spatiotemporal transition, and spatiotemporal interaction network.

#### 2.2.1. LISA Time Path

The LISA time path is a continuous expression of the Markov transition matrix [47,48], which describes the spatial dynamics and spatiotemporal synergistic changes of local variables, realizes the pairwise migration of standardized values and spatial lags of variables, which transforms the static local spatial dependence into dynamic continuous expression [47,49], and can be used to reflect the characteristics of local spatiotemporal coordinated changes of rural poverty in China.

Unit *y* in the Moran scatter plot of the transfer path is a set of vectors ((*z_y_*,_1_, *zL_y_*,_1_), (*z_y_*,_2_, *zL_y_*,_2_), …, (*z_y_,_t_*, *zL_y_,_t_*)), where *z_y_,_t_* is the normalized value of rural energy poverty of unit *y* in year *t*, and *zL_y_,t* is the spatial lag of unit *y* in year *t*. The LISA time path geometry features include the relative length (Γy) and tortuosity (*ey*). These parameters are calculated as follows [50]:$$\mathrm{Relative}\;\mathrm{length}:\;\Gamma y=\frac{m\times \sum_{t=1}^{T-1}\phi \left(L_{y,t},L_{y,t+1}\right)}{\sum_{y=1}^{m}\sum_{t=1}^{T-1}\phi \left(L_{y,t},L_{y,t+1}\right)};\;\mathrm{Tortuosity}:\;ey=\frac{\sum_{t=1}^{T-1}\phi \left(L_{y,t},L_{y,t+1}\right)}{\phi \left(L_{y,t},L_{y,T}\right)}$$
where *L_y,t_* is the LISA coordinate of the y unit at time *t; φ* (*L_y,t_, L_y,t_*_+1_) is the movement distance of unit y from time *t* to *t +* 1; T is the annual time interval; and m is the number of space units. Γy > 1 indicates that the relative length of unit *y* is greater than the average length, and that it has a more dynamic spatial structure and local spatial dependencies; the larger the *e_y_*, the more curved the path, and the more the rural energy poverty of unit *y* is affected by the neighborhood spatial effect, indicating a more dynamic local spatial dependence and a more volatile rural energy poverty growth process.

#### 2.2.2. Spatiotemporal Transition

The spatiotemporal transition can reveal the transfer characteristics of local spatial association types in different periods in a Moran scatter plot [47]. In this paper, the spatiotemporal transition was used to describe the transfer characteristics and evolution process of the local spatial association types of rural poverty in China, and can be divided into four types (Table 2) [47,49].

#### 2.2.3. Spatiotemporal Interaction Visualization

The pattern of the spatiotemporal interaction of socioeconomic attributes can also be illustrated by graph theory. This interaction helps to reveal some insignificant associations in traditional ESDA. The spatiotemporal network of rural energy poverty distribution can be visualized through the geographical center connection between provinces. Generally, the intensity of the temporal correlation between neighboring provinces is distinguished by the shape, color, and thickness of the line, so as to reveal the competition and cooperation situation among provinces in the evolution process of rural energy poverty [48]. STARS is an open integrated software designed for spatiotemporal panel data calculation. It is an exploratory spatiotemporal data analysis tool that can provide visual and dynamic spatiotemporal correlation measurement and display. In this study, the spatiotemporal network of rural energy poverty among provinces was constructed using STARS.

### 2.3. Geographical Detector

The geographical detector is a set of statistical methods to detect spatially stratified heterogeneity and reveal the driving forces behind it [51]. One of the most popular geo-detector applications is that it cannot only detect the influence of a single factor on geographical things, but can also detect the interaction between the influencing factors [52]. In addition, compared with traditional regression models, the geographic detector model calculates the effects of the explanatory variables separately, and does not need to consider the multicollinearity among them [53]. The factor explanatory power of the geo-detector is measured by the *q* value, and its expression is as follows:$$q=1-\frac{1}{N\sigma^{2}}\overset{L}{\underset{h=1}{\sum }}N_{h}\sigma_{h}^{2}$$
where *L* refers to the strata of variable *Y* (rural energy poverty) or factor *X*; *N_h_* and $\sigma_{h}^{2}$ are the number of units and variance of strata *h*, respectively; and *N* and *σ*^2^ are the total number of units and variance, respectively. The value range of *q* is [0, 1], with a higher value indicating a stronger explanatory power of the factor on rural energy poverty.

Energy poverty is a complex and multifaceted problem, which is influenced by regional economic development, income status, education level, energy investment, and management [6,24,25,28,29,30,31,54]. Based on the geographic detector model, this study took the energy poverty index as the dependent variable, and the economic development level (*GDPPC*), income level of rural residents (*INC*), education level of rural residents (*ERL*), energy investment level (*IFA*), and energy management level (*RA*) as explanatory variables (Table 1), to study the socioeconomic determinants of energy poverty in China.

## 3. Results

### 3.1. Rural Energy Poverty in China

From 2000 to 2015, China’s rural energy poverty had obvious volatility, and the overall trend was “rising first and then declining”. It can be divided into two stages. The first stage was from 2000 to 2010, during which time the energy poverty index of China’s rural areas increased from 0.761 to 0.857; the second stage was from 2010 to 2015, when the rural energy poverty index showed a steady downward trend from 0.857 to 0.791 (Figure 1).

From the perspective of the three regions, the evolution trend of the energy poverty index was basically consistent with that of the whole country during 2000–2015. The degree of rural energy poverty in eastern China has always been lower than the national average level, whereas that in central and western China has always been higher than the national average level, forming a “central–west–east” stepwise decreasing pattern. The main reason is that the rural economy in eastern China is relatively developed, and farmers have a relatively high education level and a strong sense of energy conservation and environmental protection. In addition, China has completed the west-to-east gas transmission project, the guaranteed degree of modern energy supplies has been greatly enhanced, the level of modern energy access and service is high, and the energy poverty index has been maintained at a low level. The economy of the central and western rural areas is relatively backward, traditional biomass energy and fossil fuels such as coal and firewood are the main energy sources used in rural areas, and the access and service level of modern energy are low, leading to a high degree of rural energy poverty in central and western China.

It is worth noting that in 2008, the government increased investment in rural energy construction, encouraged the development and utilization of renewable energy, such as biogas and ecological energy, significantly increased the proportion of renewable energy, and effectively suppressed rural energy poverty. In addition, China’s rural energy poverty was alleviated after 2010, mainly due to the continuous promotion of the new urbanization strategy and targeted poverty alleviation strategy, the increase of investment in rural energy infrastructure construction, the enhancement of the comprehensive supply capacity of rural energy, and the improvement of the general service level of power [6]. In addition, local governments actively encourage and guide rural residents to use clean energy such as wind and solar energy, resulting in a tendency towards reduction in the degree of rural energy poverty. However, China’s rural energy poverty continues to show a significant overall imbalance, and there is a spatial polarization phenomenon of rural energy poverty. For example, rural energy poverty in Xinjiang, Inner Mongolia, Shaanxi, Shanxi, and northeast China remains higher than the national average level.

### 3.2. Spatiotemporal Interaction of Energy Poverty in China

#### 3.2.1. Time Path Change of Rural Energy Poverty

The spatial distributions of the LISA time path of rural energy poverty in China are shown in Figure 2.

The LISA time path relative length of rural energy poverty shows that there were 14 provinces with a relative length > 1 in 2000–2015, accounting for 46.67% of the total, i.e., the local pattern of rural energy poverty exhibited a significant evolutionary law over time. During this period, the relative length showed a decreasing trend from east and west to the middle (Figure 2a), and the relative lengths of Shanghai, Zhejiang, Fujian, Guangdong, and Hainan in eastern China, and Heilongjiang in northeast of China were all > 1.25, indicating strong dynamics in their local spatial structure. Conversely, the provinces with relatively short lengths were mainly concentrated in Henan, Shanxi, Anhui, Hunan, and Shaanxi in central and western China, and Hebei in eastern China; the relative length was <0.75, indicating that the stability of the local spatial structure was the strongest. The main reason was that, in these provinces, traditional biomass energy and fossil fuels are the main energy sources used in rural life, modern energy access and service levels are low, and rural energy poverty is severe, forming a relatively stable local spatial structure of rural energy poverty.

Figure 2b shows that the tortuosity of the rural energy poverty was >1, and that the overall spatial distribution presented a decreasing change towards the surrounding areas, with Chongqing, Hubei, and Hunan showing high values, indicating that the rural energy poverty index has had a strong spatial dependence over time. In 2000–2015, provinces with high tortuosity were distributed in Hubei and Hunan in central China and Chongqing in western China, reflecting the obvious dynamic changes and interaction processes between these provinces and their neighbors. The provinces with low tortuosity were mainly distributed in Jilin, Liaoning, Henan, Anhui, Jiangsu, Shanghai, Zhejiang, Fujian, Jiangxi, and Hainan; rural energy poverty among these provinces showed a steady change direction and spatial dependence.

#### 3.2.2. Spatiotemporal Transition of Rural Energy Poverty

Spatiotemporal transition can reflect the mutual transfer of local spatial correlation types in the coordinates of rural energy poverty in China. Therefore, we utilized the spatiotemporal transition probability matrix to reveal the transfer characteristics and evolution process of local spatial correlation types of energy poverty (Table 3).

In 2000–2005, 2005–2010, 2010–2015, and 2000–2015, the proportions of type IV transition times reached 83.33%, 87.33%, 94.00%, and 88.22%, respectively, indicating obvious path dependence characteristics of the rural energy poverty distribution in China, although rural energy poverty improved after the continuous promotion of the new urbanization strategy and targeted poverty alleviation strategy. However, it is difficult to completely reverse the interaction between provinces in rural energy poverty in the short term. Among them, Xinjiang, Shanxi, Shaanxi, Guizhou, and Inner Mongolia are the provinces with severe rural energy poverty, and the stability of rural energy poverty is the key area limiting China’s coordinated energy poverty reduction. From 2000 to 2015, the number of type I and type II transitions were 25 and 24, respectively, accounting for 5.56% and 5.33%, respectively; the number of type III transitions was four, accounting for <1%. The number and proportion of the four transition types indicated that the national energy poverty reduction strategy needs to further build a collaborative mechanism of regional poverty reduction, realize the path innovation of poverty reduction under the intervention of provincial and regional external forces, and promote high-quality energy poverty reduction in severe rural energy poverty-stricken areas.

#### 3.2.3. Spatiotemporal Network of Rural Energy Poverty

To better reveal the geographical essence behind the visualization of rural energy poverty, this study depicted the spatiotemporal network of the rural energy poverty interaction in neighboring provinces (Figure 3).

From the geographical network and topological network pattern of rural energy poverty, there were 29 pairs of negative connections, accounting for 43.94%, indicating a certain degree of space–time competition in the evolution process of rural energy poverty among neighboring provinces. Among all of the negative connections, 58.62% of the provinces were mainly concentrated in central and western China. According to the theory and law of development geography, the rural energy poverty and poverty alleviation pattern among neighboring provinces were in line with the evolution process of differentiation, diffusion, and convergence. The imbalance of geographical capital space intensified the competition among provinces in energy poverty reduction, and limited the convergence of regional energy poverty reduction to a certain extent. The strong negative correlations between Shaanxi and Gansu, Inner Mongolia and Gansu, Shaanxi and Ningxia, Guangdong and Hainan, Gansu and Qinghai, Inner Mongolia and Shaanxi, Guangxi and Yunnan, etc., revealed the disharmonious characteristics of energy poverty reduction among the provinces. Each province may have certain regional closure when implementing poverty reduction measures, which is specifically reflected in the relatively weak effect of regional energy poverty reduction around the administrative boundary. These negatively related regions should pay more attention to the role of collaborative poverty reduction. On the other hand, Anhui and Henan, Inner Mongolia and Jilin, Jilin and Heilongjiang, Hebei and Shanxi, and Liaoning and Jilin constituted strong positive connection provinces, showing spatial dynamics of positive coordination. The characteristics of rural energy poverty in the above provinces were similar, and the situation of regional concentration of rural energy poverty appeared. However, it is still possible to carry out further energy poverty alleviation cooperation and promote overall energy poverty alleviation.

### 3.3. Socioeconomic Determinants of Rural Energy Poverty

#### 3.3.1. Factor Analysis of Determinants of Rural Energy Poverty

The power of determinant values (*q*) of each driving factor are shown in Figure 4. Considering the average value of the power of determinants from 2000 to 2015, it is evident that the mean *q* values for the disposable income of rural residents (0.478) was the highest, followed by GDP per capita (0.426), education level of rural labor (0.377), regulatory agency (0.228), and energy investment (0.205).

As shown in Figure 4, the disposable income of rural residents is the most important factor affecting rural energy poverty. The *q* value of the impact of the disposable income of rural residents on rural energy poverty from 2000 to 2015 was between 0.357 and 0.566. Especially after 2010, among the influencing factors, rural residents’ disposable income has had the greatest impact on energy poverty, indicating that the disposable income of rural residents is one of the most critical factors that could significantly reduce rural energy poverty. In general, people in energy poverty cannot afford to adopt low-carbon technology such as solar photovoltaic panels [55,56,57], and increasing the disposable income of rural residents will reduce the share of coal consumption and enhance the ability to access clean and stable modern energy services, thereby alleviating rural energy poverty [58,59]. Oum pointed out that if income is insufficient, access to electricity for poor households may increase the pressure on their income and expenditure, leading to energy poverty [60]. A study conducted by Lin et al. showed that increasing residents’ income is conducive to promoting residents to change their energy consumption mode, and can significantly improve rural households’ energy poverty situation [61].

GDP per capita is the fundamental driver of rural energy poverty, and contributes a remarkably prominent impact on rural energy poverty in China compared to other driving factors, generating a *q* value from 0.306–0.637. According to the energy ladder hypothesis, with the improvement in the economic development level, household energy consumption is gradually moving towards clean and efficient modern energy [62,63,64]. The hypothesis points out that economic development can alleviate energy poverty, and existing studies have provided strong evidence for this. For example, Acharya et al. investigated the relationship between energy poverty and economic development in India, and recorded a negative relationship between economic development and energy poverty [65]. Lee studied the influencing factors of household energy consumption in Uganda, and found that the proportion of electricity consumption increased with the development of the economy, and that solid fuel utilization showed an inverted U-shaped feature [66]. Arthur et al. found that relatively wealthy households in Mozambique are more likely to use electricity as their main living energy, whereas relatively poor families mostly use firewood [67].

The education level of rural labor had a *q* value ranging from 0.142 to 0.539, revealing a rather significant impact on rural energy poverty. The impact of education level on energy poverty is mainly reflected by income and substitution effects. The income effect means that the education level is generally positively correlated with the income level; therefore, the group with a lower education level often has a lower income, which affects their consumption of high-quality energy. The substitution effect refers to residents with a lower education level having more leisure time to collect traditional biomass energy sources (such as firewood and straw) because of their low work remuneration and low time cost. Meanwhile, for people with a lower education level, the willingness to use clean cooking utensils and the ownership of household appliances are relatively low; therefore, it is difficult to increase clean energy consumption. Some scholars have also proved that education level has an important impact on energy poverty. For example, Acharya et al. found that among the components of economic development, education has a greater impact on reducing energy poverty than other factors [65]. Liang et al. found that the household energy consumption structure of Zhaotong City in Yunnan Province is closely related to the per capita education level [68].

The *q* values of the regulatory agency were higher than those of the energy investment level. The regulatory agency generated a *q* value from 0.034 to 0.489, indicating that it had significantly high determinant power on rural energy poverty. In fact, the establishment and improvement of rural energy management institutions can restrain rural energy poverty. Among the influencing factors, energy investment level had the weakest influence on China’s energy poverty, generating a *q* value from 0.002 to 0.444. The *q* values of the energy investment level increased from 0.013 in 2000 to 0.225 in 2015, showing a remarkable increase and that the impacts of investment in fixed assets in the state-owned economic energy industry on rural energy poverty have gradually strengthened. With the steady improvement of social and economic development, China’s investment in energy popularization, energy infrastructure construction, renewable energy, and energy efficiency has increased significantly, and remarkable achievements have been made in the development and utilization of energy resources and infrastructure construction. According to the “global renewable energy investment trends in 2016” report released by the United Nations Environment Programme, China’s investment in renewable energy reached USD 102.9 billion in 2015, accounting for more than one third of the global investment in renewable energy. China’s rural energy poverty has been significantly improved, and the clean energy supply capacity has also increased. Therefore, we can assume that the energy investment level has an important influence on rural energy poverty.

Moreover, on closer inspection, the *q* values of *GDPPC*, *INC*, *ERL*, *IFA*, and *RA* all showed obvious decreases after 2010. Combined with the trend of average rural energy poverty in China (Figure 1), in which rural energy poverty in China was alleviated after 2010, we believe that there exist other potential factors with significant impacts on rural energy poverty. Specifically, the impacts of factors such as energy substitution and poverty alleviation policy and projects resulted in the impact of the drivers exhibiting a downward trend after 2010. For instance, since 2012, a large range of haze weather has appeared in northern China in winter. To reduce air pollution, relevant national ministries and commissions have issued a series of policies to support the comprehensive replacement of bulk coal by clean heating, with the continuous implementation of China’s “coal to gas” policy, natural gas has gradually entered the rural areas from the city, which has effectively alleviated rural energy poverty [69]. In 2014, the National Energy Administration and the Poverty Alleviation Office of the State Council jointly formulated the work plan for the implementation of the photovoltaic poverty alleviation project, implemented distributed photovoltaic poverty alleviation, and installed a distributed photovoltaic power generation system, to increase the basic living conditions of people on low incomes and to suppress energy poverty.

#### 3.3.2. Interaction Analysis of Determinants of Rural Energy Poverty

This paper not only identifies the separate effects of the drivers of rural energy poverty in China, but also further examines the interaction relationships between them. Table 4 shows the *q* values for the interactive effects between drivers in 2000, 2005, 2010, and 2015. In Table 4, the *q* values on the diagonal line refer to the separate effects of each driver as shown in Figure 4, and the lower triangular matrix shows the *q* values of the interactive effects between the drivers. The interaction results revealed that interactions between five driving factors showed bi-enhanced interactive and nonlinear enhanced effects on rural energy poverty in China.

In 2000, a part of the bi-enhanced interactive effects was observed between the driving factors, e.g., *GDPPC* and *INC* (0.732), *GDPPC* and *ERL* (0.840), *GDPPC* and *RA* (0.710), *INC* and *ERL* (0.600), and *ERL* and *RA* (0.535), a situation that indicates more interactive effects relative to the separate effects from any single driving factor. In addition, *GDPPC* and *IFA* (0.749), *INC* and *IFA* (0.740), *INC* and *RA* (0.733), *ERL* and *IFA* (0.506), and *IFA* and *RA* (0.534) showed nonlinear enhanced effects, that is, interactive effects between these drivers exceeded the sum of separate effects. In 2005, *GDPPC* and *INC* (0.557), *GDPPC* and *ERL* (0.649), *GDPPC* and *IFA* (0.717), *INC* and *ERL* (0.631), *INC* and *IFA* (0.664), and *ERL* and *IFA* (0.892) showed bi-enhanced interactive effects, whereas the other influencing factors showed nonlinear enhanced effects. In 2010, *GDPPC* and *IFA* (0.725) showed nonlinear enhanced effects, whereas the other influencing factors showed bi-enhanced interactive effects. In 2015, *GDPPC* and *INC* (0.626), *INC* and *IFA* (0.691), *INC* and *RA* (0.641), and *ERL* and *IFA* (0.421) showed bi-enhanced interactive effects, and the other influencing factors showed nonlinear enhanced effects. Overall, our findings showed that all five driving factors had enhanced effects on rural energy poverty in China through interaction effects.

## 4. Conclusions, Policy Implications, and Limitations

### 4.1. Conclusions

This study investigated the spatiotemporal interaction characteristics of rural energy poverty using the ESTDA framework, and then applied the geographical detector to quantify the socioeconomic determinants of rural energy poverty. The main results are as follows.

(1)From 2000 to 2015, China’s rural energy poverty had obvious volatility, and the overall trend was “rising first and then declining”. The evolution trend of the energy poverty of the three regions tended to be consistent with that of the whole country, and formed a “central–west–east” stepwise decreasing pattern. Since 2010, China’s rural energy poverty has been alleviated; however, it continues to show a significant overall imbalance, and there is a spatial polarization phenomenon of regional rural energy poverty.(2)There was a dynamic local spatial dependence and a volatile rural energy poverty evolution process in China, and the spatial agglomeration of energy poverty had obvious path dependence and locked spatial features. The provinces with negative connections were mainly concentrated in central and western China. Anhui and Henan, Inner Mongolia and Jilin, Jilin and Heilongjiang, Hebei and Shanxi, and Liaoning and Jilin constituted strong synergistic growth areas. Carrying out coordinated poverty reduction should become the focus of sustainable energy poverty reduction in the future.(3)All of the five drivers explored in this study had significant impacts on rural energy poverty. In the long run, the disposable income of rural residents had the greatest determinant power on rural energy poverty, followed by GDP per capita, education level of rural labor, regulatory agency, and energy investment. In addition, the interaction results revealed that interactions between all explanatory variables showed bi-enhanced interactive and nonlinear enhanced effects on rural energy poverty in China.

### 4.2. Policy Implications

The results show that the spatial agglomeration of rural energy poverty in China has high path dependence and locking spatial characteristics, and is affected by many factors. Therefore, to solve the problem of rural energy poverty in China more effectively, it is urgent to formulate and implement differentiated policies according to the actual situation of rural energy access and energy services in various provinces.

For provinces with high rural energy poverty, such as Xinjiang, Shaanxi, and Chongqing, the government should develop the rural economy, increase the disposable income of residents, and improve the ability to pay for clean energy. In addition, the government should increase investment in rural energy, strengthen the construction of rural energy management and promotion institutions, further improve the construction of rural power infrastructure, deepen the popularity of rural power services, and increase the investment in rural clean energy development and related infrastructure construction. In particular, it is necessary to give full play to the advantages of clean energy in the western region, vigorously develop wind, solar, geothermal, and other clean energy, and improve the level and completeness of clean energy infrastructure. At the farmers level, it is necessary to strengthen education and publicity, raise public awareness of and advocate a green lifestyle, and adopt modern, clean, and efficient energy utilization. At the same time, it is necessary to actively use modern life energy consumption equipment to improve the popularization rate of modern life energy consumption equipment actively.

In addition, it is necessary to build a cooperative mechanism for cross-regional energy poverty reduction, actively play the leading role of pilot demonstration, encourage regions with low energy poverty to take the lead in demonstration, and use spatial autocorrelation to produce positive spillover effects and positive externalities.

### 4.3. Limitations and Prospects

Our research mainly focused on the spatiotemporal interaction characteristics and main socioeconomic determinants of rural energy poverty at the provincial level in China. However, the strategies to alleviate energy poverty will ultimately be implemented at the micro scale, such as county, village, and households; therefore, the investigation on rural energy poverty at the micro scale should be strengthened in the future. Moreover, this study only explored the situation of rural energy poverty in China from 2000 to 2015, and the development of renewable energy in rural China has made great progress since 2015. Determining whether the transformation from fossil fuels to renewable energy in rural China has alleviated energy poverty requires further study. Finally, geographical conditions and policy factors are important factors affecting rural energy poverty, which need to be strengthened in the follow-up study.

## Figures and Tables

**Figure 1 ijerph-19-10851-f001:**
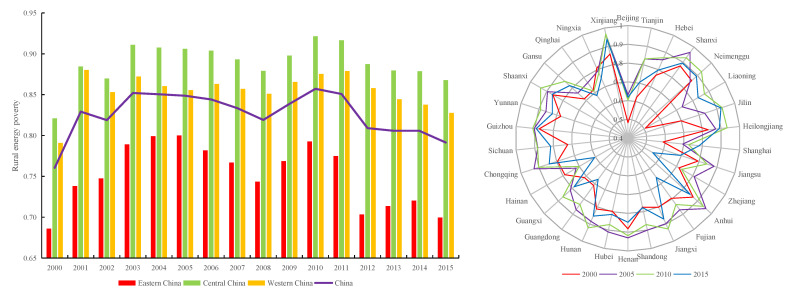
Rural energy poverty in China, 2000–2015. Note: Eastern China: Beijing, Tianjin, Hebei, Liaoning, Shanghai, Jiangsu, Zhejiang, Fujian, Shandong, Guangdong, and Hainan; Central China: Shanxi, Jilin, Heilongjiang, Anhui, Jiangxi, Henan, Hubei, and Hunan; Western China: Inner Mongolia, Chongqing, Sichuan, Guizhou, Yunnan, Tibet, Shaanxi, Gansu, Qinghai, Ningxia, Xinjiang, and Guangxi.

**Figure 2 ijerph-19-10851-f002:**
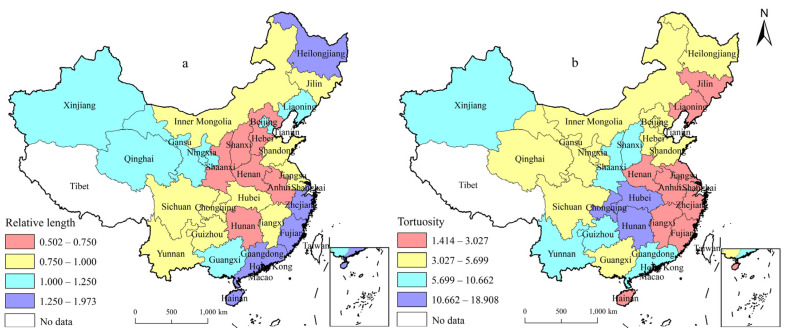
Spatial distributions of the (**a**) relative length and (**b**) tortuosity of rural energy poverty.

**Figure 3 ijerph-19-10851-f003:**
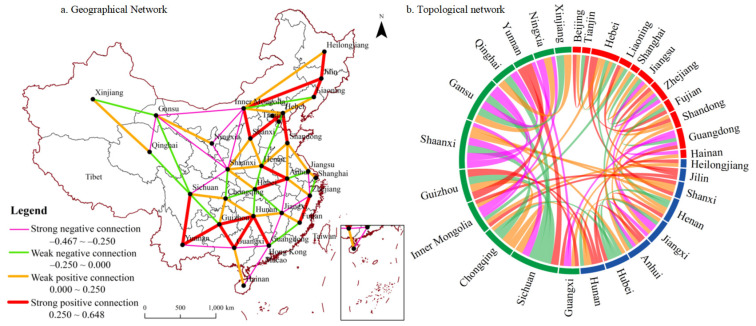
Spatiotemporal network of rural energy poverty interaction among provinces in China. (**a**) Geographical network; (**b**) topological network.

**Figure 4 ijerph-19-10851-f004:**
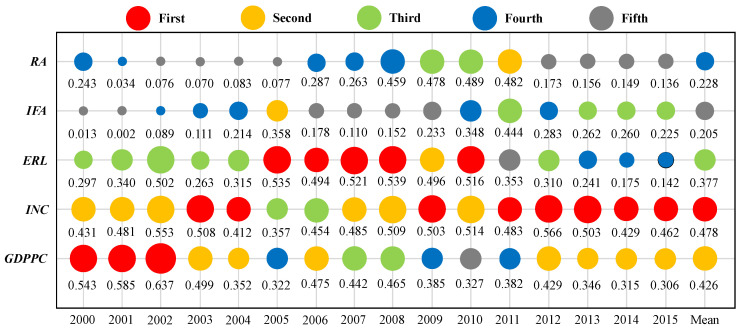
Power of determinant value (*q*) of each driving factor from 2000 to 2015. *RA*—energy management level, *IFA*—energy investment level, *ERL*—education level of rural residents, *INC*—income level of rural residents, *GDPPC*—economic development level. Note: colors indicate the rank of the factor force, and the circle size indicates the level of influence of the factor force.

**Table 1 ijerph-19-10851-t001:** Details of data used in this study.

Data Type	Data Name	Data Description	Source
Rural energy poverty	Energy access	Per capita domestic electricity consumption (kwh), per capita liquefied petroleum gas (kg), per capita total gas output of biogas digester (m^3^), per capita solar water heater (m^3^), per capita solar room (m^3^)	*China Rural Statistical Yearbook* (2001–2016), *China Rural Energy Yearbook* (2001–2016), *China Energy Statistics Yearbook* (2001–2016)
	Energy service	Clean cookware penetration rate (kitchen ventilator (one for every 100 households), solar cooker (one for every 100 households))	
Socioeconomic factors	Economic development level	GDP per capita (*GDPPC*) (yuan)	*China Statistical Yearbook* (2001–2016)
	Income level of rural residents	Disposable income of rural residents (*INC*) (yuan)	*China Rural Statistical Yearbook* (2001–2016)
	Education level of residents	Education level of rural labor (*ERL*) (year/person)	
	Energy investment level	Investment in fixed assets in state-owned economic energy industry (*IFA*) (10^8^ yuan)	
	Energy management level	Regulatory agency (*RA*) (number)	

**Table 2 ijerph-19-10851-t002:** Spatiotemporal transition type of rural energy poverty.

Type	Meaning	Transition Characteristics
Ⅰ	Only local rural energy poverty in transition	LL_t_→HL_t+1_, LH_t_→HH_t+1_, HL_t_→LL_t+1_, HH_t_→LH_t+1_
Ⅱ	Only neighborhood rural energy poverty in transition	LL_t_→LH_t+1_, LH_t_→LL_t+1_, HL_t_→HH_t+1_, HH_t_→HL_t+1_
Ⅲ	Both local and neighborhood rural energy poverty in transition	LL_t_→HH_t+1_, LH_t_→HL_t+1_, HL_t_→LH_t+1_, HH_t_→LL_t+1_
Ⅳ	The local and neighborhood rural energy poverty as stable	LL_t_→LL_t+1_, LH_t_→LH_t+1_, HL_t_→HL_t+1_, HH_t_→HH_t+1_

**Table 3 ijerph-19-10851-t003:** Spatiotemporal transition matrices of rural energy poverty.

Period of Time	t/t + 1	HH	LH	LL	HL	Type	*n*	Proportion
2000–2005	HH	0.867	0.083	0.000	0.050	Ⅰ	13	8.67%
	LH	0.139	0.722	0.139	0.000	Ⅱ	12	8.00%
	LL	0.000	0.040	0.920	0.040	Ⅲ	0	0.00%
	HL	0.103	0.000	0.069	0.828	Ⅳ	125	83.33%
2005–2010	HH	0.958	0.000	0.000	0.042	Ⅰ	7	4.67%
	LH	0.042	0.875	0.083	0.000	Ⅱ	10	6.67%
	LL	0.053	0.079	0.816	0.053	Ⅲ	2	1.33%
	HL	0.118	0.000	0.235	0.647	Ⅳ	131	87.33%
2010–2015	HH	0.973	0.014	0.000	0.014	Ⅰ	5	3.33%
	LH	0.125	0.813	0.000	0.063	Ⅱ	2	1.33%
	LL	0.000	0.023	0.977	0.000	Ⅲ	2	1.33%
	HL	0.000	0.059	0.118	0.824	Ⅳ	141	94.00%
2000–2015	HH	0.937	0.029	0.000	0.034	Ⅰ	25	5.56%
	LH	0.105	0.789	0.092	0.013	Ⅱ	24	5.33%
	LL	0.019	0.047	0.906	0.028	Ⅲ	4	0.89%
	HL	0.079	0.016	0.127	0.778	Ⅳ	397	88.22%

**Table 4 ijerph-19-10851-t004:** Interaction of driving factors on rural energy poverty in 2000, 2005, 2010, and 2015.

2000						2005					
	*GDPPC*	*INC*	*ERL*	*IFA*	*RA*		*GDPPC*	*INC*	*ERL*	*IFA*	*RA*
*GDPPC*	0.543					*GDPPC*	0.322				
*INC*	0.732	0.431				*INC*	0.557	0.357			
*ERL*	0.840	0.600	0.297			*ERL*	0.649	0.631	0.535		
*IFA*	0.749	0.740	0.506	0.013		*IFA*	0.717	0.664	0.892	0.358	
*RA*	0.710	0.733	0.535	0.534	0.243	*RA*	0.684	0.671	0.744	0.595	0.077
**2010**						**2015**					
	*GDPPC*	*INC*	*ERL*	*IFA*	*RA*		*GDPPC*	*INC*	*ERL*	*IFA*	*RA*
*GDPPC*	0.327					*GDPPC*	0.306				
*INC*	0.684	0.514				*INC*	0.626	0.462			
*ERL*	0.831	0.702	0.516			*ERL*	0.518	0.722	0.142		
*IFA*	0.725	0.785	0.810	0.348		*IFA*	0.618	0.691	0.421	0.225	
*RA*	0.747	0.897	0.850	0.875	0.489	*RA*	0.575	0.641	0.570	0.488	0.136
		Bi-enhanced				Nonlinear-enhanced			Separate effects		

## Data Availability

We obtained the data from *Journal of Global Change Data & Discovery*, China Statistical Yearbook 2001–2016, China Rural Statistical Yearbook 2001–2016.

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
