# Peer review of "Spatiotemporal Interaction and Socioeconomic Determinants of Rural Energy Poverty in China"

_ijerph, 2022, doi:10.3390/ijerph191710851_

Round 1

Reviewer 1 Report

This paper explored the energy poverty spatio-temporal interaction characteristics and socio-economic determinants in rural China from 2000-2015. The conclusion has some guiding significance for reducing the rural energy poverty. In addition, energy poverty issue has attracted considerable attention from scholars in recent years, this research has certain practical significance and necessity. However, there are several issues that need further improvement.

1) In general, the introduction is quite comprehensive. It is suggested to reduce the first and second paragraphs, and the literature review is more focused on the spatial and temporal distribution of rural energy poverty and its influencing factors.

2)  At present, the necessity and significance of the paper is not clear enough, so it is suggested to improve. The contribution to this paper needs to be further strengthened.

3) The reasonableness of the indicator selection needs to be explained. Why are geographical conditions or policy factors not taken into account in the indicator.

4) In section 3.1, it seems that rural energy poverty was low in 2008. Please explain why?

5) In section 4.2, it is suggested that the author put forward targeted opinions and suggestions on the influencing factors of rurals' energy poverty mentioned above.

Author Response

Reviewer 1

This paper explored the energy poverty spatio-temporal interaction characteristics and socio-economic determinants in rural China from 2000-2015. The conclusion has some guiding significance for reducing the rural energy poverty. In addition, energy poverty issue has attracted considerable attention from scholars in recent years, this research has certain practical significance and necessity. However, there are several issues that need further improvement.

Point 1: In general, the introduction is quite comprehensive. It is suggested to reduce the first and second paragraphs, and the literature review is more focused on the spatial and temporal distribution of rural energy poverty and its influencing factors.

Response 1: Thanks for reviewer’s suggestion. We have reduced the first and second paragraphs, and the literature review focuses more on the spatial and temporal differentiation and influencing factors of energy poverty. Please refer to the introduction.

Point 2: At present, the necessity and significance of the paper is not clear enough, so it is suggested to improve. The contribution to this paper needs to be further strengthened.

Response 2: Thanks for reviewer’s suggestion. We have clarified the possible contribution of this paper in detail, that is, although a great number of studies have been conducted on energy poverty, two limitations are still existed: First, some previous studies have investigated the spatial pattern of energy poverty by exploratory spatial data analysis (ESDA); however, ESDA only targets cross-sectional data and ignores the temporal and spatial dynamics of rural energy poverty. In fact, rural energy poverty is a complex process of space‒time change, which will vary with changes in time and space. This article introduces the exploratory time‒space data analysis (ESTDA) framework to study the spatio‒temporal interaction characteristics of rural energy poverty in China. Importantly, the ESTDA framework can integrate time and space to study spatio‒temporal interaction as well as to compensate for the shortcomings of ESDA, which ensures the accuracy of the estimation results. Second, previous studies have generally focused on the impact of single factor, such as household income, energy prices, buildings, and equipment on energy poverty. However, rural energy poverty is complex and may be affected by many factors. It is urgent to explore the interaction of multiple factors on energy poverty. Therefore, this study involved a single factor and interactive analysis on the socio-economic factors of rural energy poverty to better understand the spatio‒temporal dynamics and socio-economic determinants of rural energy poverty in China. Please refer to lines 56-73.

Point 3: The reasonableness of the indicator selection needs to be explained. Why are geographical conditions or policy factors not taken into account in the indicator.

Response 3: Thanks for reviewer’s suggestion. The indicators selected in this paper, such as the disposable income of rural residents, per capita GDP, rural labor education level, regulatory agencies, and energy investment, have been used in existing studies, and relevant literature has been cited as support in this paper, so we will not repeat them in this paper. See lines 175-182 for details. This paper does not consider geographical conditions and policy factors, mainly for the following two reasons. First of all, this paper takes the provincial units as the research scale. This research scale is large, and it is not significant to consider the geographical conditions. Secondly, policy factors are difficult to quantify. Considering the availability of data, this paper does not consider the impact of policy factors on rural energy poverty in China. This deficiency has been pointed out in detail in the limitations and prospects. See lines 496-497.

Point 4: In section 3.1, it seems that rural energy poverty was low in 2008. Please explain why?

Response 4: Thanks for reviewer’s suggestion. We have explained why China’s rural energy poverty was relatively low in 2008. In 2008, the government increased investment in rural energy construction, encouraged the development and utilization of renewable energy, such as biogas and ecological energy, significantly increased the proportion of renewable energy, and effectively suppressed rural energy poverty. Please refer to lines 212-215.

Point 5: In section 4.2, it is suggested that the author put forward targeted opinions and suggestions on the influencing factors of rurals’ energy poverty mentioned above.

Response 5: We thank the reviewer’s helpful comments and strongly agree with. Based on research results, we improved the policy recommendations.

The results show that the spatial agglomeration of rural energy poverty in China has high path dependence and locking spatial characteristics, and is affected by many factors. Therefore, to solve the problem of rural energy poverty in China more effectively, it is urgent to formulate and implement differentiated policies according to the actual situation of rural energy access and energy services in various provinces.

For provinces with high rural energy poverty, such as Xinjiang, Shaanxi and Chongqing, the government should develop the rural economy, increase the disposable income of residents and improve the ability to pay for clean energy. In addition, the government should increase investment in rural energy, strengthen the construction of rural energy management and promotion institutions, further improve the construction of rural power infrastructure, deepen the popularity of rural power services, and increase the investment in rural clean energy development and related infrastructure construction. In particular, it is necessary to give full play to the advantages of clean energy in the western region, vigorously develop wind, solar, geothermal, and other clean energy, and improve the level and completeness of clean energy infrastructure. At the farmers level, it is necessary to strengthen education and publicity, raise public awareness and advocate a green lifestyle, adopt modern, clean, and efficient energy utilization. At the same time, it is necessary to actively use modern life energy consumption equipment, to improve the popularization rate of modern life energy consumption equipment actively.

In addition, it is necessary to build a cooperative mechanism for cross-regional energy poverty reduction, actively play the leading role of pilot demonstration, encourage regions with low energy poverty to take the lead in demonstration, and use spatial autocorrelation to produce positive spillover effects and positive externalities. Please refer to lines 462-485.

Reviewer 2 Report

In this study, energy poverty in rural China was examined in terms of its spatial and temporal characteristics and socioeconomic determinants. Overall, it is an interesting study. But after reading the paper, I have several questions. 

  1. The methodology is not very clear to me. Several methods are listed in the paper, but I had difficulties following each method. For example, line 134 introduced an "explained variable", and lines 135-137 listed "explanatory variables". It sounds like an econometric model is used, but instead of discussing the econometric model, it is followed by the ESTDA and LISA. I don't understand whether the ESTDA and LISA are methods developed and used in this study, or just indexes to measure energy poverty.    2. How about the multicollinearity among the independent variables? There could be strong multicollinearity among income, GDPPC, and energy investment levels.    3. I don't see the relationship between the results and the policy implications.    4. Why did the authors separate the analysis into different time periods?    5. Table 1 lists different variables under "Energy access". But it is not clear to me how energy poverty was quantified in this study.    I think overall, the topic is interesting, but I don't see a clear contribution to the literature. There is a strong relationship between energy poverty and income/GDP. As income changes unevenly, energy poverty is not evenly distributed temporally and spatially. It sounds like a straightforward statement to me. So the contributions of this study are not clear to me. 

Author Response

Reviewer 2

In this study, energy poverty in rural China was examined in terms of its spatial and temporal characteristics and socioeconomic determinants. Overall, it is an interesting study. But after reading the paper, I have several questions.

Point 1: The methodology is not very clear to me. Several methods are listed in the paper, but I had difficulties following each method. For example, line 134 introduced an “explained variable”, and lines 135-137 listed “explanatory variables”. It sounds like an econometric model is used, but instead of discussing the econometric model, it is followed by the ESTDA and LISA. I don’t understand whether the ESTDA and LISA are methods developed and used in this study, or just indexes to measure energy poverty.

Response 1: Thanks for reviewer’s suggestion. This manuscript uses the exploratory time-space data analysis (ESTDA) framework to analyze the spatio-temporal interaction characteristics of rural energy poverty in China. The ESTDA analysis framework including the local indicators of spatial association (LISA) time path, spatio-temporal transition, and spatio-temporal interaction network. Please refer to lines 116-119. This manuscript investigated the socio-economic determinants of rural energy poverty in China using geographical detector model, line 134 introduced an “explained variable”, and lines 135-137 listed “explanatory variables” for geographical detector model. Please refer to lines 175-182. The rural energy poverty index used in this paper was released by the Journal of Global Change Data & Discovery [46]. Please refer to lines 105-106.

Point 2: How about the multicollinearity among the independent variables? There could be strong multicollinearity among income, GDPPC, and energy investment levels.

Response 2: Thanks for reviewer’s suggestion. There is no doubt that many socio-economic factors affecting rural energy poverty may have multicollinearity. This manuscript investigated the socio-economic determinants of rural energy poverty in China using geographical detector model. Studies have shown that compared with traditional regression models, the geographic detector model calculates the effects of the explanatory variables separately, and does not need to consider the multicollinearity among them. Please refer to lines 166-168. Therefore, this study did not calculate the multicollinearity among the independent variables.

Point 3: I don’t see the relationship between the results and the policy implications.

Response 3: We thank the reviewer’s helpful comments and strongly agree with. Based on research results, we specified the policy recommendations.

The results show that the spatial agglomeration of rural energy poverty in China has high path dependence and locking spatial characteristics, and is affected by many factors. Therefore, to solve the problem of rural energy poverty in China more effectively, it is urgent to formulate and implement differentiated policies according to the actual situation of rural energy access and energy services in various provinces.

For provinces with high rural energy poverty, such as Xinjiang, Shaanxi and Chongqing, the government should develop the rural economy, increase the disposable income of residents and improve the ability to pay for clean energy. In addition, the government should increase investment in rural energy, strengthen the construction of rural energy management and promotion institutions, further improve the construction of rural power infrastructure, deepen the popularity of rural power services, and increase the investment in rural clean energy development and related infrastructure construction. In particular, it is necessary to give full play to the advantages of clean energy in the western region, vigorously develop wind, solar, geothermal, and other clean energy, and improve the level and completeness of clean energy infrastructure. At the farmers level, it is necessary to strengthen education and publicity, raise public awareness and advocate a green lifestyle, adopt modern, clean, and efficient energy utilization. At the same time, it is necessary to actively use modern life energy consumption equipment, to improve the popularization rate of modern life energy consumption equipment actively.

In addition, it is necessary to build a cooperative mechanism for cross-regional energy poverty reduction, actively play the leading role of pilot demonstration, encourage regions with low energy poverty to take the lead in demonstration, and use spatial autocorrelation to produce positive spillover effects and positive externalities. Please refer to lines 462-485.

Point 4: Why did the authors separate the analysis into different time periods?

Response 4: Thanks to the suggestions of the reviewers, this paper divides the analysis process into different time periods, mainly for the following two considerations. Firstly, it can be seen from Figure 1 that China’s rural energy poverty has the characteristics of stage and regional differentiation, and the analysis of different time periods can more comprehensively reveal the spatial and temporal differentiation characteristics of China’s rural energy poverty. Secondly, the analysis process is divided into different time periods to facilitate the comparison of the characteristics of China’s energy poverty in different stages.

Point 5: Table 1 lists different variables under “Energy access”. But it is not clear to me how energy poverty was quantified in this study.

Response 5: The rural energy poverty index used in this paper was released by the Journal of Global Change Data & Discovery [46]. The index was constructed by Zhao et al. from the two dimensions of energy access and energy services (Table 1), was calculated by the weighted summation method [6], and can comprehensively reflect the situation of energy poverty in rural China. This study used this index to study the spatio‒temporal interaction characteristics and socio-economic determinants of rural energy poverty in China. Please refer to lines 105-111.

Point 6: I think overall, the topic is interesting, but I don’t see a clear contribution to the literature. There is a strong relationship between energy poverty and income/GDP. As income changes unevenly, energy poverty is not evenly distributed temporally and spatially. It sounds like a straightforward statement to me. So the contributions of this study are not clear to me.

Response 6: Thanks to the suggestions of the reviewers. We have clarified the possible contribution of this paper in detail, that is, although a great number of studies have been conducted on energy poverty, two limitations are still existed: First, some previous studies have investigated the spatial pattern of energy poverty by exploratory spatial data analysis (ESDA); however, ESDA only targets cross-sectional data and ignores the temporal and spatial dynamics of rural energy poverty. In fact, rural energy poverty is a complex process of space‒time change, which will vary with changes in time and space. This article introduces the exploratory time‒space data analysis (ESTDA) framework to study the spatio‒temporal interaction characteristics of rural energy poverty in China. Importantly, the ESTDA framework can integrate time and space to study spatio‒temporal interaction as well as to compensate for the shortcomings of ESDA, which ensures the accuracy of the estimation results. Second, previous studies have generally focused on the impact of single factor, such as household income, energy prices, buildings, and equipment on energy poverty. However, rural energy poverty is complex and may be affected by many factors. It is urgent to explore the interaction of multiple factors on energy poverty. Therefore, this study involved a single factor and interactive analysis on the socio-economic factors of rural energy poverty to better understand the spatio‒temporal dynamics and socio-economic determinants of rural energy poverty in China. Please refer to lines 56-73.

Reviewer 3 Report

This paper presents a significant contribution to the field, with an average originality and quality of presentation. The scientific background and foundations are solid and it will be of interest to the readers. The study allows for a good overall vision on rural energy poverty in China and the results are clearly presented.

Howevere, as a possible contribution of this paper you underline the introduction of the exploratory time‒space data analysis (ESTDA) framework to study the spatio‒temporal interaction characteristics of rural energy poverty in China to compensate the shortcomings of ESDA and to ensure the accuracy of the estimation results. It would be necessary that you clarify the differences between your analysis framework and the one used in the paper published in 2018, [8]: Zhao, X. Y.; Chen, H. H.; Ma, Y. Y.; Gao, Z. Y.; Xue, B: Spatio-temporal variation and its influencing factors of rural energy 517 poverty in China from 2000 to 2015. Geographical Research 2018, 37 (6), 1115-1126. doi: 10.11821/dlyj201806005.

Unfortunately, the .doi link does not work so it is impossible to assess the relation with the present work.

Author Response

Reviewer 3

This paper presents a significant contribution to the field, with an average originality and quality of presentation. The scientific background and foundations are solid and it will be of interest to the readers. The study allows for a good overall vision on rural energy poverty in China and the results are clearly presented.

Howevere, as a possible contribution of this paper you underline the introduction of the exploratory time‒space data analysis (ESTDA) framework to study the spatio‒temporal interaction characteristics of rural energy poverty in China to compensate the shortcomings of ESDA and to ensure the accuracy of the estimation results. It would be necessary that you clarify the differences between your analysis framework and the one used in the paper published in 2018, [8] Zhao, X. Y.; Chen, H. H.; Ma, Y. Y.; Gao, Z. Y.; Xue, B: Spatio-temporal variation and its influencing factors of rural energy poverty in China from 2000 to 2015. Geographical Research 2018, 37 (6), 1115-1126. doi: 10.11821/dlyj201806005.

Unfortunately, the doi link does not work so it is impossible to assess the relation with the present work.

Response: Thanks for reviewer’s suggestion. This paper, titled “Spatio-temporal variation and its influencing factors of rural energy poverty in China from 2000 to 2015”, through an integration of Theil index, exploratory spatial data analysis (ESDA) and spatial panel econometric model to examine the spatio-temporal patterns and influencing factors of rural energy poverty in China. In contrast to this paper, this manuscript investigated the energy poverty spatio-temporal interaction characteristics and socio-economic determinants in rural China from 2000‒2015 using exploratory time-space data analysis (ESTDA) and geographical detector model. ESDA only targets cross-sectional data and ignores the temporal and spatial dynamics of rural energy poverty. In fact, rural energy poverty is a complex process of space‒time change, which will vary with changes in time and space. This article introduces the exploratory time‒space data analysis (ESTDA) framework to study the spatio‒temporal interaction characteristics of rural energy poverty in China. Importantly, the ESTDA framework can integrate time and space to study spatio‒temporal interaction as well as to compensate for the shortcomings of ESDA, which ensures the accuracy of the estimation results. Second, previous studies have generally focused on the impact of single factor, such as household income, energy prices, buildings, and equipment on energy poverty. However, rural energy poverty is complex and may be affected by many factors. It is urgent to explore the interaction of multiple factors on energy poverty. Therefore, this study involved a single factor and interactive analysis on the socio-economic factors of rural energy poverty to better understand the spatio‒temporal dynamics and socio-economic determinants of rural energy poverty in China. Please refer to lines 56-73.